# Inter-Organizational Supply Chain Interaction for Sustainability: A Systematic Literature Review

**Veronica S Ülgen [1], Maria Björklund [2,\*], Niklas Simm [2]**  **and Helena Forslund [1]**

1   Department of Management Accounting and Logistics, Linnaeus University, 35195 Vaxjo, Sweden;
    veronica.svensson.ulgen@lnu.se (V.S.U.); helena.forslund@lnu.se (H.F.)
2   Department of Logistics and Quality Management, Linköping University, 58183 Linkoping, Sweden;
    niklas.simm@liu.se
\*   Correspondence: maria.bjorklund@liu.se

**Abstract:** Research on inter-organizational supply chain interaction (IOSCI) for sustainability can be described as fast-growing and fragmented. It is uncertain to what extent logistics and supply chain management (SCM) literature has been able to close research gaps regarding e.g., empirical scope and sustainability dimensions addressed. A systematic literature review (SLR) is carried out to synthesize the existing knowledge and ongoing trends, and to propose a research agenda. The trend analysis shows that the gap between the environmental and social dimension is increasing, that Asian studies grow nearly exponentially, and that the focal firm perspective gains more ground. The research agenda calls for participatory studies of multiple actors and for studies addressing, e.g., the invisible logistics service providers. This study provides an important complement to existing literature reviews on sustainable supply chain interaction, primarily by its focus on the development over time, the empirical scope, the focus on publications in logistics/SCM journals, and its narrow focus on the interaction among firms in supply chains.

**Keywords:** inter-organizational; collaboration; co-ordination; supply chains; logistics; systematic literature review

## 1. Introduction

This review is a highly extended version of a former conference paper [1]. The focus of sustainability research and firm practice in the 1990s and early 2000s was the sustainable behavior and alignment of the economic, environmental, and social dimensions on a firm level [2,3]. According to Krause et al., [4], a firm is not more sustainable than its supply chain. The development towards sustainable firms is dependent on other actors in the chain [5–7]. Firms are recognizing the need to address sustainability in a way that also encompasses the operations of other actors in the supply chain [8]. Practices performed in firm-internal isolation are said to be less successful [7] and firms at the end of the supply chain often find themselves held responsible for the entire chain [8]. To not move sustainability problems between supply chain actors, inter-organizational interaction is called for. Interaction can range from strategic, based on collaboration and/or partnership, e.g., [9,10], to more operational forms of cooperation, e.g., [11,12].

Research on inter-organizational interaction for sustainability (or similar terms) is fragmented, focusing on environmental [13–15], social [16] or triple bottom line [17–19] dimensions of sustainability and addresses different actors such as retailers [20], third-party logistics service providers [21] or manufacturers [22]. It also investigates different effects of IOSCI, such as effects on quality [23], innovation and knowledge sharing [24], implementation of sustainable supply chain policies [25], and reputation and legitimacy [26]. As a response to the fragmented knowledge, the area of sustainable

supply chain interactions has been targeted by structured literature reviews (SLR). These studies focus the collaboration with actors within and outside the supply chain [27], or the mapping of theory use in SSCM in operations and supply and business ethics/sustainability journals [28]. The SLR by Touboulic and Walker [28] is one of several studies within the logistics/SCM field emphasizing the need for more research targeting social sustainability as the environmental dimension is clearly overrepresented. It is unclear to what extent later research has been able to narrow this gap or if the gap is expanding.

Despite the emphasis on the importance of interaction between multiple actors in supply chains [29,30]; calls for research with data collection from several actors [31,32] and supply chains as the unit of analysis, it is unclear to what extent contemporary research has been able to narrow this gap, or if the dyadic or focal firm perspective is growing even stronger. One explanation to the identified limited scope could be the focus on more strategic supply chain interaction in contemporary literature reviews, e.g., [27,33], as relations with 2nd and 3rd tier suppliers or customers are seldom strategic. This calls for a review of existing literature applying a more holistic view of different types of inter-organizational interaction (e.g., collaboration, coordination, and cooperation) between actors in supply chains, here termed inter-organizational supply chain interaction (IOSCI). Inter-organizational interaction for sustainability is addressed from different disciplines such as operations/production management [27,34,35], business ethics [36,37], environmental management [38], and logistics [39]. SCM and logistics journals should be able to capture research with data collection from several supply chain actors, and it is this literature that is currently under scrutiny. Logistics, including transportation are known to have significant sustainability impacts [21,40]. Unravelling the developments over time for previously identified research gaps, as well as unravelling the developments in other areas can give an understanding of where the research on inter-organizational interaction is, but also where it is heading. This enables identification of potential future research gaps.

The purpose of this article is to advance the understanding of the logistics and supply chain management literature on IOSCI for sustainability by describing and synthesizing the existing knowledge, the development of the research over time, and to propose a future research agenda.

This study has an important complementing focus to existing SLRs by: (1) applying the wider view on interaction—including strategic to more operational types; (2) focusing on interaction between firms within supply chains; (3) identification of ongoing trends, providing not only a snap-shot of the status-addressing whether previously identified gaps are narrowing or increasing, and whether it is possible to identify potential future gaps. This is an important point of departure when identifying future relevant research paths. Critically evaluating extant research and identifying new directions and research questions worthy of investigation are important aspects for advancing knowledge in the field. The review is guided by the following research questions:

*RQ1. In what ways has research on IOSCI for sustainability in the supply chain management literature developed over time?*

*RQ2. In what ways has the research on IOSCI for sustainability managed to employ a wide research scope?*

*RQ3. What are the most important aspects to address in a future research agenda in this area?*

The remainder of the article is structured as follows: first is the presentation of the SLR methodology, followed by a description and synthesis of the investigated literature. The remaining sections present the identification of research gaps and finally the proposed research agenda.

## 2. Methods

A systematic literature review (SLR) was performed. The SLR "locates existing studies, selects and evaluates contributions, analyses and synthesizes data, and reports the evidence in such a way that allows reasonably clear conclusions to be reached about what is and is not known" [41] (p. 671). The SLR is suitable for identifying and summarizing common areas in a field, contrasting the differences, assessing the existing intellectual territory, identifying research gaps, and providing concrete propositions for a future research agenda [41,42]. Replicability is enabled when performing

a systematic literature review [43], as it implies detailed descriptions of all steps taken to identify, scan, and analyze the literature. This increases transparency and reduces biases [31,42]. The reviewer team, consisting of four authors, collaborated and had joint discussions in all steps of this literature review, however also elements of independent work were included.

*The Research Process*

The research process began with defining the topic, the unit of analysis and the formulation of research questions (see RQ1–3 in the Introduction) to guide the review, in line with recommendations of Denyer and Tranfield [41] and Durach et al., [44]. The UoA is articles in logistics and supply chain management journals addressing interactions for sustainability between firms in supply chains. Based on the identified research focus IOSCI for sustainability, and in line with earlier SLRs in the research domain, e.g., [27,45–47], the keywords used to locate the studies were then identified. Two categories of keywords were applied: (1) keywords capturing sustainability (environment*, sustainab*, "corporate responsib*", csr, "social responsib*", "socially responsib*", green, "triple bottom line" and 3bl*), and (2) words to capture the inter-organizational/collaborative dimensions (collaborat*, co-ordinat*, co-operat* and inter-organization*, with different spellings). The "*" sign at the end of keywords, was used to widen the range of studies, and the keywords used were quite broad (e.g., including not only environment but also green) to ensure that studies adopting alternative vocabulary were identified. "Interaction" however, is too general a term, therefore not used as a keyword. The search string was based on all combinations possible between the two categories, using the Boolean connector "OR" internally within the category of keywords and "AND" between the categories, inspired by, e.g., Pilbeam et al., [48]. To ensure high quality, only peer-reviewed journal articles were included [28], which is a common practice in logistics [31] and sustainability research, e.g., [45,49]. The search was performed in Scopus and in Web of Science (WoS). A field-competent librarian was consulted for the selection of databases, declaring the Scopus database as having the best coverage in the field and WoS as the best complementary database. This approach was confirmed by a recently published SLR [50]. With the aim to synthesize the knowledge in logistics and supply chain management, as opposed to, e.g., production/operations management or business ethics, the search was limited to journals with a clear and narrow focus on logistics and supply chain management. It is relevant to target not only sustainability journals; rather to bring in knowledge to sustainability journals from other areas. No limitations regarding year of publication was applied. The search (performed 24 August, 2018) was conducted in title, abstract, and keywords, inspired by the approach applied by Wong et al., [51] and Björklund and Johansson [52]. When the list was cleared from duplicates, 627 articles were identified, where WoS added 106 to the 521 identified in Scopus.

The screening process started with the scanning of abstracts based on the parameters: (1) inter-organizational supply chain interaction, and (2) sustainability for inclusion/exclusion purposes. Related to sustainability, environmental and/or social aspects were focused. Economic sustainability was not an inclusion criterion on its own as this dimension is already present in the research within the logistics and SCM field. For calibration purposes, the first twenty abstracts were read by all authors. Any deviations in classification were discussed between the authors, and consensus to either include or exclude was reached. The remaining 608 abstracts were consecutively read. Guided by the recommendations of Tranfield et al., [42], at least two authors reviewed each abstract to avoid subjective decisions. Five reviewer pairs were used to identify potential biases. The inter-rater agreement of the initial classification was 90%. Discrepancies were discussed and reconciled, and when needed, a third author was involved. The review gave 119 articles for inclusion as they matched the two parameters of IOSCI and sustainability.

Articles were excluded based on the following criteria: (1) wrong type of sustainability (e.g., sustainable competitive advantage) or environment (e.g., competitive or learning environment); (2) interaction with other actors than those within the same supply chain (e.g., government or competitors); (3) wrong type of interaction (e.g., coordination between green-lights). More than 200 articles were excluded based on the first

criteria, and over 260 based on the second. A smaller group (40) was excluded based on the third criteria. As the exclusion criteria were not mutually exclusive several articles could have been excluded based on more than one criterion. Seven articles were excluded due to difficulties in accessing the full text version of the article. The complete reading of the 112 articles done by a minimum of two authors, excluded 25 additional articles by the use of a fourth exclusion criteria (4); articles not focusing on the interaction or sustainability dimensions addressed, thus not increasing the understanding related to the purpose of the study. 87 articles covering IOSCI for sustainability remained for the final analysis. Figure 1 illustrates the research process.

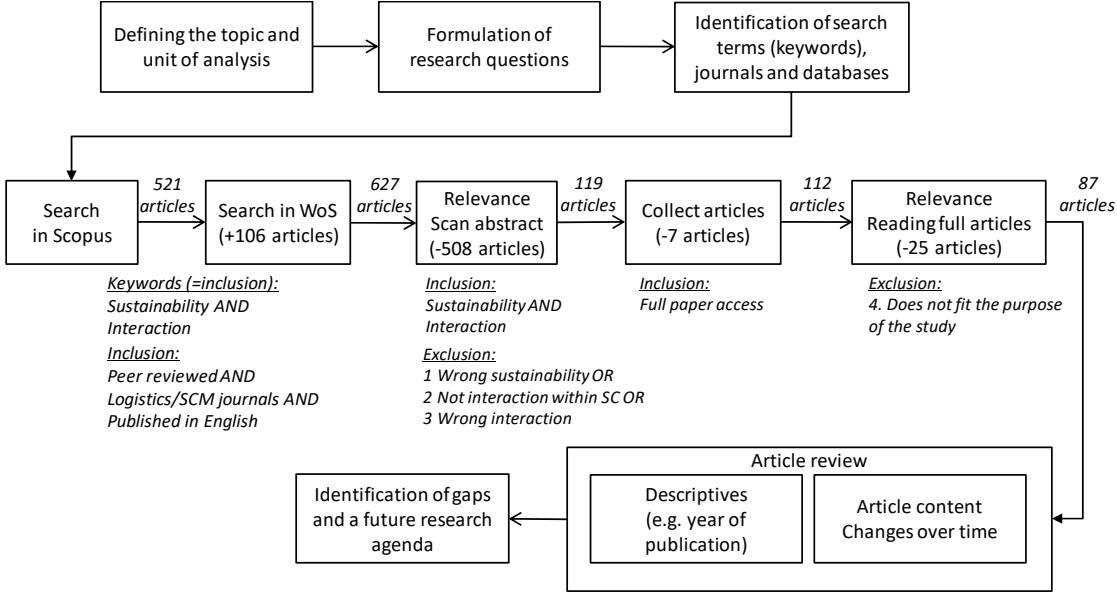

**Figure 1.** Illustration of the research process.

Inspired by Tranfield et al., [42] an initial spreadsheet data-extraction form, including general (title, author, and publication details) and descriptive information (e.g., methods) was created. The data-extraction form with the coding manual, i.e., the instructions to support the coding [53], was further developed and fine-tuned during the complete article reading. The selected articles are presented in Table 1, below, and marked with * in the reference list, in line with a previous SLR [54].

**Table 1.** Coding of the 87 articles included and marked with * in the reference list.

| | ID for Included Articles | | | | |
|---|---|---|---|---|---|
| [55] | Chabot et al. (2018) | [56] | Azar et al. (2018) | [57] | Venturini et al. (2017) |
| [58] | Xie et al. (2017) | [59] | Hyder et al. (2017) | [60] | Chen et al. (2017) |
| [10] | Murfield and Tate (2017) | [39] | Liljestrand (2017) | [61] | Soysal (2016) |
| [20] | Pålsson and Hellström (2016) | [62] | Sallnäs (2016) | [63] | Miemczyk et al. (2016) |
| [64] | Touboulic and Walker (2015) | [30] | Wiengarten and Longoni (2015) | [65] | Jabbour et al. (2015) |
| [66] | Hsueh (2015) | [11] | De Giovanni and Vinzi (2014) | [67] | So and Xu (2014) |
| [68] | Beske and Seuring (2014) | [69] | Panda (2014) | [14] | Theißen et al. (2014) |
| [70] | Blanquart and Carbone (2014) | [71] | Golicic and Smith (2013) | [72] | Wong (2013) |
| [73] | Brockhaus et al. (2013) | [43] | Miemczyk et al. (2012) | [74] | Shi et al. (2012) |
| [75] | Kirchoff et al. (2011) | [76] | Peters et al. (2011) | [77] | Alvarez et al. (2010) |
| [78] | Bansal and Mcknight (2009) | [79] | Maon et al. (2009) | [12] | Ciliberti et al. (2009) |
| [80] | Cheng et al. (2008) | [81] | Rodrigues et al. (2008) | [82] | Lee (2008) |
| [83] | Vanek and Morlok (2000) | [84] | Knight et al. (2017) | [85] | Shen et al. (2017) |

**Table 1.** *Cont.*

| | ID for Included Articles | | | | |
|---|---|---|---|---|---|
| [86] | Hajmohammad and Vachon (2016) | [87] | Lion et al. (2016) | [88] | Modak et al. (2016) |
| [89] | Nair et al. (2016) | [90] | Rodríguez et al. (2016) | [91] | Xu et al. (2016) |
| [92] | Chen et al. (2015) | [9] | Luo et al. (2015) | [93] | Koppius et al. (2014) |
| [29] | Meehan and Bryde (2014) | [94] | Cheng (2011) | [95] | Gavronski et al. (2011) |
| [96] | Large and Thomsen (2011) | [97] | Glock et al. (2012) | [98] | Liu (2018) |
| [99] | Wolf and Seuring (2010) | [100] | Hall and Matos (2010) | [101] | Tate et al. (2014) |
| [102] | Sundram et al. (2017) | [103] | Prasad et al. (2017) | [104] | Schneider and Wallenburg (2012) |
| [105] | Caniëls et al. (2013) | [106] | Hoejmose et al. (2014) | [107] | Yawar and Kauppi (2018) |
| [108] | Busse (2016) | [17] | Ni and Sun (2018) | [109] | Bai et al. (2017) |
| [110] | Green et al. (2012) | [111] | Pedersen (2009) | [15] | Yang (2018) |
| [112] | Zissis et al. (2018) | [113] | Sheu (2011) | [114] | Hafezalkotob (2017) |
| [115] | Carter and Jennings (2002) | [116] | Biswas et al. (2018) | [117] | Stevenson and Cole (2018) |
| [118] | Konur (2017) | [16] | Normann et al. (2017) | [26] | Ali et al. (2017) |
| [119] | Tachizawa and Wong (2015) | [32] | Soosay and Hyland (2015) | [120] | Gold et al. (2015) |
| [22] | Yu et al. (2014) | [23] | Gualandri et al. (2014) | [121] | Moxham and Kauppi (2014) |
| [122] | Hoejmose et al. (2013) | [123] | Crespin-Mazet and Dontenwill (2012) | [124] | Xie and Breen (2012) |

The research process is in several ways in line with the modified AMSTAR criteria [125] in terms of á priori defining inclusion criteria, clearly defining and presenting keywords, searching in multiple databases, using minimum two researchers in article selection and in the review of full papers for content analysis, and presenting the list of included papers (Table 1). The type of literature included is explicitly stated, however based on Touboulic, and Walker [28] narrowed to only peer reviewed journals.

## 3. Results

Guided by e.g., Manders et al. [126], the findings are based on an analysis of the source and date of publication, unit of analysis, research methods applied, industry and geographical area, as well as the theoretical approach. After the presentation of sources and dates of publication this chapter commences with a description of the sustainability dimensions identified and their developments over time. Thereafter an elaboration on the unit of analysis, empirical scope and methodology of the included papers is presented. Unit of analysis, empirical scope and methodology is used for further analysis in the coming sections (industry and geographical area, theoretical lens and finally type of sustainability interaction). The inclusion of empirical scope for data collection adds an additional nuance regarding the interactions for sustainability studied.

### 3.1. Source and Date of Publication

The 87 articles identified are distributed over 13 logistics, including transportation, and SCM journals. A nearly exponential increase in publications can be viewed since the first article in 2000, with more than 25% of the articles published in 2017 and 2018 (see Figure 2). This development is in line with the general increase in publications within the logistics and supply chain fields, yet illustrating the importance granted to sustainability research within the logistics and SCM areas.

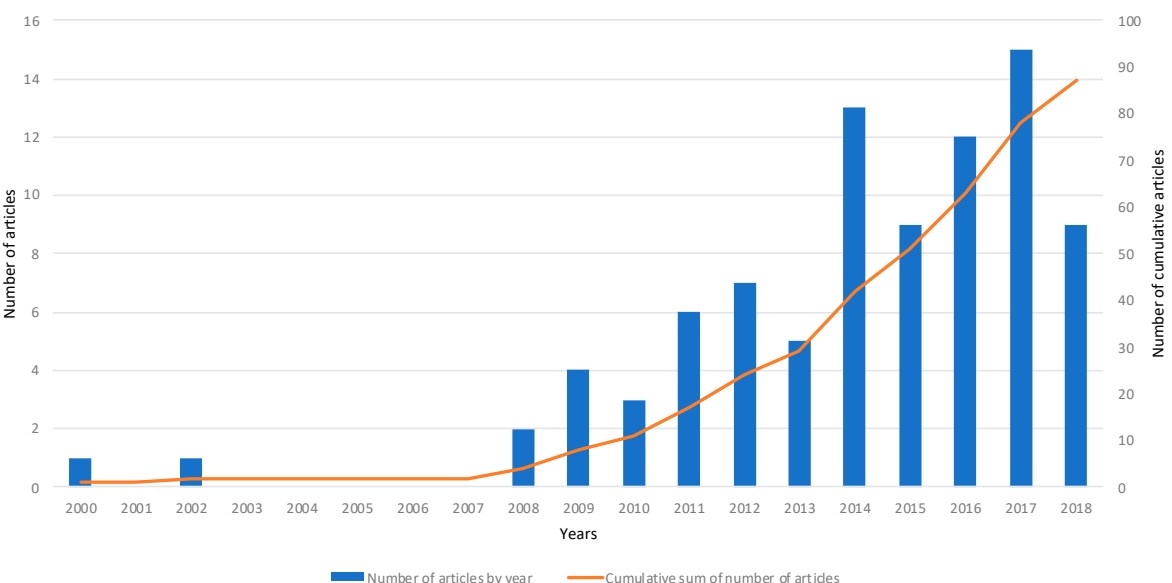

**Figure 2.** Number of articles by year, and cumulative sum of articles by year.

### 3.2. Sustainability Dimension

The sampled articles show an overweight of the environmental dimension compared to the social dimension (see Figure 3), in line with other reviews within sustainable logistics and SCM [28,31]. The trend analysis adds one important insight: not only is there a domination of the environmental dimension, it also grows faster over time compared with the social dimension. As discussions on climate change is a constant companion in society at large, and environmental issues like the Volkswagen emission scandal takes place, the focus onto the environmental dimension can be readily explained.

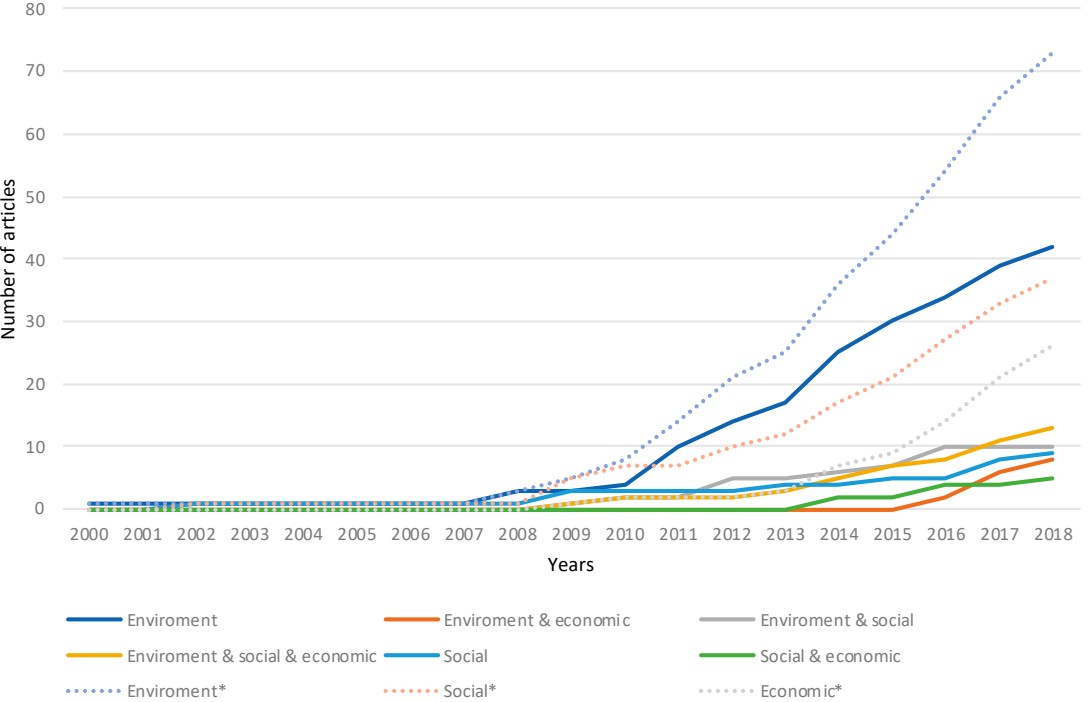

**Figure 3.** Development of "sustainability dimension" in articles over time. *Notes*: * The sum of articles including the sustainability dimension with or without other dimensions.

Within the environmental dimension a broadening outside the expected focus on $CO_2$ [11,30,64] is identified. Beyond the broad "climate related category" focusing ($CO_2$) emissions [11,14,64,83,97], and waste and waste reduction [68,78,97,126] the environmental dimension can be seen as fragmented that covers various aspects. The social dimension regards primarily codes of conduct and/or working conditions at suppliers and sub-suppliers in developing countries in Asia [12,16,59,103,106]. Empirical evidence of socially focused IOSCI that include suppliers outside of Asia or service suppliers are almost absent. Another finding is the focus on e.g., CSR-performance [66,69] or green supply chain collaboration [9,56] where the sustainability aspects are generally held, and presented in terms of e.g., environment-sustaining activities. Hence, both specific (e.g., waste reduction) and rather vague (e.g., 3BL) sustainability aspects are focused in the sustainable logistics and SCM-research. Based on our limited sample it is not possible to draw any conclusions on trends related to what aspects of environmental or social sustainability have been in focus at different times.

The economic dimension is by nature embedded in most logistics and SCM research; therefore, the classification of the economic dimension was based on articles that clearly elaborated on the connection to this dimension. A recent and quickly growing trend (starting around year 2016) is the consideration of the combined environmental and economic performance. One explanation is that the environmental dimension is no longer seen as an economic burden, rather a business opportunity with a positive effect on performance based on the understanding that the dimensions are interlinked [127]. Several studies center the effect on performance. In addition to the articles focusing on social and environmental performance, the performance effects relate to the economic dimension, through improved quality [30], increased flexibility [30], profit [11,26,66], reduced costs [70], improved financial outcome [55,91], decreased risks [10], innovativeness [72], competitive advantages [63,73,92,97], green value creation [70], positive reputation and legitimacy [26,63], and operational performance [23,30,71]. Improved performance is however described in unspecific terms in most studies—where few show actual measurements for the performance effects, or capture investments or costs—however, there are exceptions with a clearer focus on performance measurement [71,96]. Furthermore, negative performance effects are also shown where collaboration for sustainability may lead to increased cost [96]. Oftentimes, the research is clearer in terms of the aspects in focus within the environmental dimension than within the other two dimensions, which could possibly be explained by the business related research area.

### 3.3. Unit of Analysis and Empirical Scope

The unit of analysis (UoA) in the sampled articles, was investigated from an IOSCI point of view. Three UoA are identified: dyad, supply chain, and network. Dyad is represented by the IOSCI between two supply chain actors; supply chain is represented by IOSCI between a minimum of three actors in different positions in the supply chain (e.g., manufacturer, wholesaler, retailer), and network is represented by IOSCI between three or more supply chain actors with interaction "one to many" (see Figure 5 for clarification).

The dyad [10,56,61,99,109] and the supply chain [20,39,73,98,106] are the most common units of analysis (see Figure 4), represented by studies of, for example, environmental concerns in buyer- (LSP)- supplier dyads [99] and environmental efficiency of packaging from a manufacturer-distributor-retailer perspective [20]. At a much slower growth rate, as well as amount, are the articles with the UoA equal to network [77,123], where for example governance mechanisms for sustainability in OEM-traders-farmers networks are addressed [77]. However, the importance of interaction beyond the dyadic relationship to address sustainability challenges is stressed within the sample [29,30,32,43]. Studies of networks complement supply chain studies as they may address different research questions and represent other types of complexities through indirect connections between relationships [128]. Those indirect relationships may make e.g., survey studies more complicated which is reflected in the sample where no survey study with UoA equal to network was identified. Also, in order to study interaction for sustainability in true networks, the needed access to e.g., customer or supplier records, and additional contact information might be difficult to attain; which might explain why few such studies are presented.

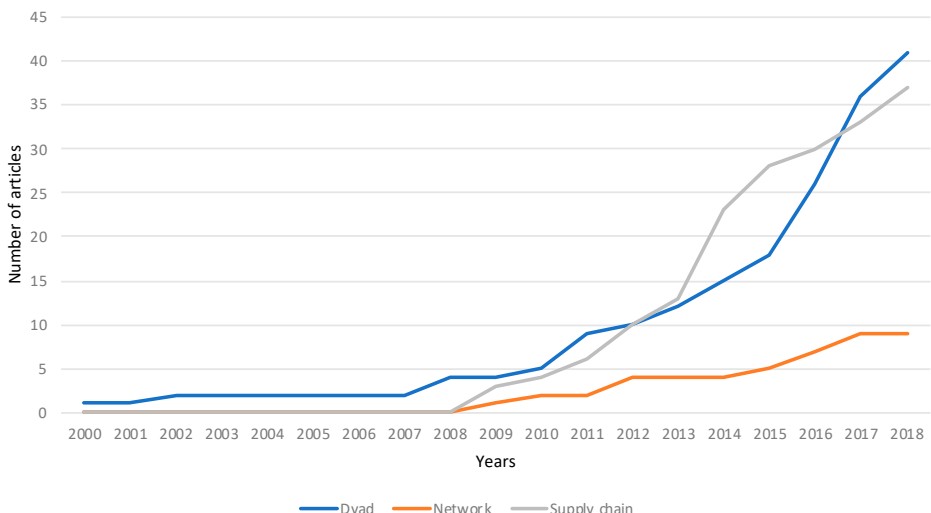

**Figure 4.** Development of "Unit of analysis" in articles over time.

A complementary analysis applicable for empirical articles with data collection on IOSCI for sustainability is the investigation of the actor perspective and where, or with how many actors, data collection takes place. This is referred to as "empirical scope." Eight empirical scopes are identified. The first three are the "true" dyads [62,99], supply chains [20,39,70] or networks [59,77] where data is collected from all actors involved in the investigated IOSCI. The next three are the "focal firm" dyads [16,56,82,91,105], supply chains [12,15,65,106] or networks [103,123] where data is collected from fewer instances in the dyad/supply chain/network than stated in the UoA. The last two identified empirical scopes are the "not interlinked" dyads [10] and supply chains [22,73], with data collection from all instances in the UoA. However, the data is not from the interlinked actors in an actual dyad or supply chain. A visualization of units of analysis and empirical scopes in the studied articles is found in Figure 5. The data collection (at what actors) and what IOSCI is investigated is illustrated through the "eye" and the direction of the view.

**Figure 5.** Unit of analysis (UoA) and exemplifying the empirical scope in the studied articles. (Number of articles within brackets, n.e = non empirical, e = empirical).

Most of the articles has a focal firm empirical scope (see Figure 5). This is in line with the findings of Kembro and Näslund [129]. Unexpectedly, the articles with true supply chains are more common than those with true dyads. Furthermore, a small possible trend can be seen indicating a lower growth rate regarding the true dyads and supply chains, as opposed to those with a focal firm perspective.

Since five of the empirical scopes are represented by only a few articles—between one and five—they have been clustered in order to allow for continued analysis. The clustering was made to clarify the different empirical scopes based on the number of actors; the three clusters are:

- (1) actor: Focal firm dyad, focal firm supply chain, and focal firm network (35 articles)

- (2) actors: True dyad and dyads not interlinked (6 articles)
- (3) (or more) actors: True supply chains, true network, and supply chains not interlinked (15 articles)

The most dominant clustered empirical scope (data collection at one actor) also has the fastest growing trend (see Figure 6). Also, studies with data collection from three or more actors display a clearly positive trend.

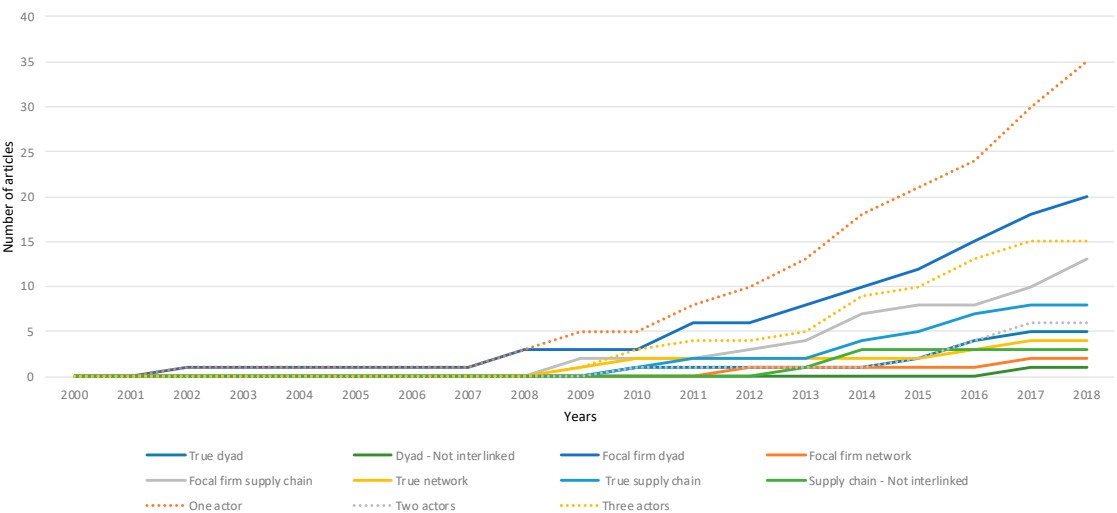

**Figure 6.** Development of "Empirical scope" in articles over time.

### 3.4. Methods Applied

The most common data collection method applied is the survey. Within the group of surveys, the favored methods for analysis are structural equation modeling [11,56,65,94,105] and regression analysis [29,82,92,102,106,122] applied to determine e.g., relationships between green practices and operational performance, or competitiveness of a firm [92,102]. An example of sustainability measures included in a survey [91] is the suppliers' environmental performance based on common measures such as carbon emission and waste reduction, together with more rarely investigated aspects such as consumption of hazardous materials and environmental accidents. The second most common data collection method is the interview. Figure 7 indicates a potential small shift around year 2015 in favor of interviews and mathematical modelling compared to survey methods. The aspects of the sustainability dimensions in the modeling articles [58,66,69] often remain unknown; instead, more general terms such as "environmental or CSR performance" is used and modeled, however not without exceptions as displayed by [83] centering efficiency in transportation. A majority of the interview-based publications are framed as qualitative case studies [16,62,64,87,99,107], combined with data obtained from observations [39], internal documents/firm web pages [85], governmental documents [100], focus groups [100], press publications [70], and quantitative data from annual reports [85]. This indicates that the logistics/SCM journals do not display the same research gap as the operations/production management journals with regards to combining diversified methodologies [27]. The case studies are primarily multiple case studies [20,39,62,87,107], in line with recommendations of Yin [130]. An example of a case study finding is that despite an increased supplier interest in sustainability, the buyer's sourcing criteria are still the traditional performance objectives [99]. The cases have been analyzed through a within- and cross-case analysis. One case study stands out as it has an interpretive approach and applies grounded theory [10], and manages to demonstrate how collaboratively developed environmental performance metrics support the implementation of environmental initiatives and increase the strategic importance of the buyer- supplier relationship. Others stand out as longitudinal single case studies [77,89,123], and/or employing action research

methodology [123] allowing for the identification of how the supply network and sourcing practices have evolved over time based on a sustainability strategy. Empirical research is most common, and our sample indicates that publications with empirical data have a stronger growth rate than the non-empirical publications. The third most common method is literature review. The literature review on sustainable purchasing [43] also provides a list of sustainability aspects addressed regarding sustainable purchasing: material use, waste, pollution, energy, and climate gases (environmental aspects), and social equity, support communities, (non) ethical behavior and compliance (social aspects).

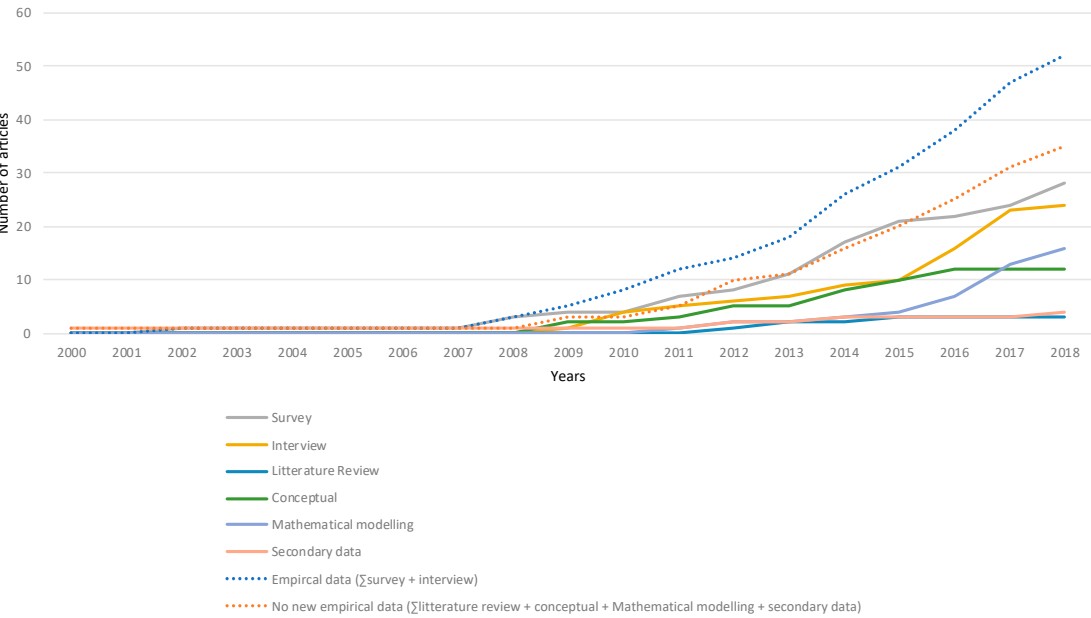

**Figure 7.** Development of "Methods applied" in articles over time.

Most of the studies with a focal firm (one actor) empirical scope are survey studies (see Table 2) where one type of supply chain actor, e.g., manufacturers, has been addressed with questions regarding the IOSCI with customers and/or suppliers [17,65,72,91]. These articles often take a buyer firm's perspective, which has previously been shown to be most common in research of buyer-supplier interaction [131,132] (see Table 2). The case methodology is largely represented in the studies of true dyads, supply chains or networks, i.e., studies with an empirical scope covering two, three, or more actors in the supply chain [39,59,70,77,89,99]. Approximately half of the non-empirical articles have the dyad as unit of analysis.

**Table 2.** Clustered empirical scope in relation to Method, as well as areas elaborated on later in the article (Geographical area, Theoretical lens, and Sustainability dimension.

| Empirical Scope * | Method | | | | | | Geographical Area | | | | | | Theoretical Lens | | Sustainability Dimension | | |
|---|---|---|---|---|---|---|---|---|---|---|---|---|---|---|---|---|---|
| | Survey | Interview | Litterature review | Conceptual | Mathematical modelling | Secondary data | Asia | Europé | Several continents | North America | South America | No theory | Theory | Enviroment ** | Social ** | Economic ** |
| One actor | 25 | 8 | 0 | 1 | 1 | 0 | 15 | 11 | 1 | 5 | 1 | 20 | 15 | 29 | 15 | 8 |
| Two actors | 0 | 6 | 0 | 0 | 0 | 0 | 1 | 2 | 1 | 1 | 1 | 3 | 3 | 5 | 1 | 1 |
| Three actors | 3 | 10 | 0 | 1 | 0 | 1 | 4 | 4 | 2 | 2 | 1 | 8 | 7 | 14 | 7 | 3 |

Notes: * Empirical scopes clustered after the number of actors studied. ** The sum of articles including the sustainability dimension with or without other dimensions.

### 3.5. Industry and Geographical Area

In terms of industries studied, single industries such as textiles [63,85,103,117], foods [26,64,77], and automotive [105,122] are focused on. The textiles industry research can be illustrated by investigating the detection and remediation of modern slavery through interaction based on monitoring of the supply chain [117]. The largest category includes more than one industry, primarily survey studies of different manufacturers [23,30,56,110], and multiple case studies [14,59,89] within different industries. An investigation of addressed supply chain actors displayed that—as in the very definition of a supply chain—actors such as suppliers [12,64,77,87,117], manufacturers [12,22,30,56,59,82], retailers/brands [20,26,59,73,88], and wholesalers, e.g., [39] are included; even if studies of wholesalers are rare. Empirical data on IOSCI from the logistics service provider (LSP) is rare [133]; presented in only four studies [62,70,73,99], despite the transport sector being the second largest emitter of green-house gases [134].

The geographical area is important when selecting cases, as differing cultures and legislation would affect the chosen sustainability initiatives [14]. Most of the empirical data originates from Asia [30,72,82,98,107] and Europe [14,39,64,111], while few studies have data from North [84,110] and South America [65] (see Figure 8). No studies from Australia/Oceania and Africa were identified. The strong growth in Asian studies might be explained by the ongoing outsourcing to Asia as well as the increase of journal publications by Asian researchers. Data collection within, e.g., global supplier networks [23,117] or from actors within different geographical areas [12,59,77] are referred to as "Several continents" in Figure 8. The few studies stretching geographical areas is somewhat surprising, again with reference to the large outsourcing or purchasing share of, e.g., European or North American manufacturers and retailers from other geographical areas. Based on the data it is not possible to propose any connections between geographical area and empirical scope (see Table 2).

The importance of studies from diverse geographical areas can be illustrated by drivers of IOSCI for sustainability. For example, factors—external to the supply chain—such as market requirements [32] and regulatory requirements [63,64,87,101,117] are seen as enablers or drivers for IOSCI, whereas in others [15,26,107] they are not. Instead, questions of legitimacy [107] and the adoption of internal green practices [15] are seen as the primary drivers. Differences have been identified between the European [106] and Asian [15,107] contexts.

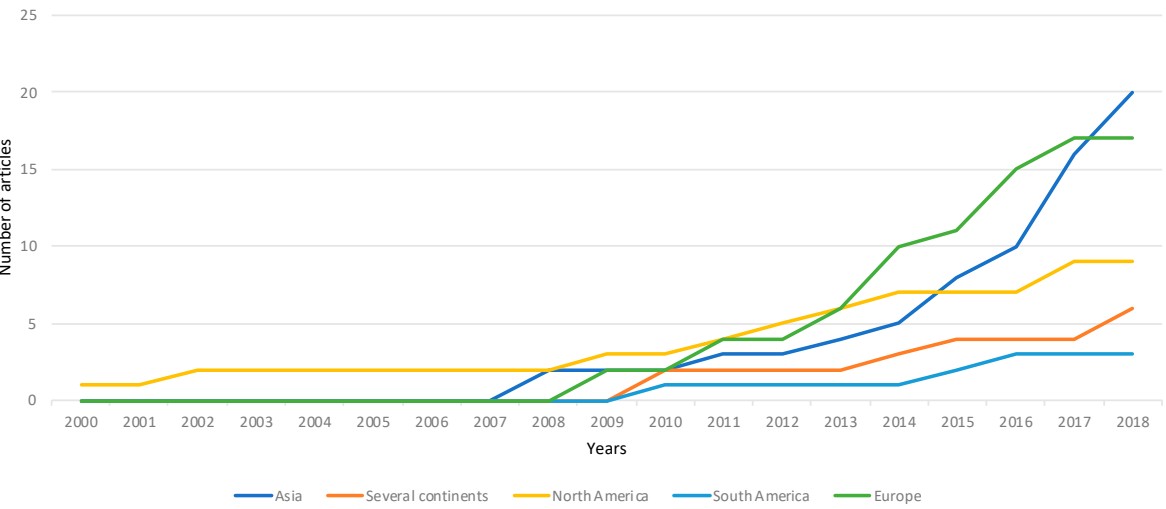

**Figure 8.** Development of "Geographical area" in articles over time.

### 3.6. Theoretical Lens

An important element in the development of any research field is the theoretical lens [135] as it represents "a way of seeing," and therefore may sharpen the inquiry of organizational life [136]. Theoretical lenses are applied in almost half of the sampled articles. This is a higher share than the

findings reported by Touboulic and Walker, [28] in their mapping of theory use in SSCM. A positive trend can be seen in the number of articles applying a theoretical lens and in the 2013–2014 gap between non-theoretical and theoretical articles narrowed (see Figure 9). Increased theory use is one important prerequisite for increased contribution to theory, a research need within sustainable supply chain management previously suggested [28].

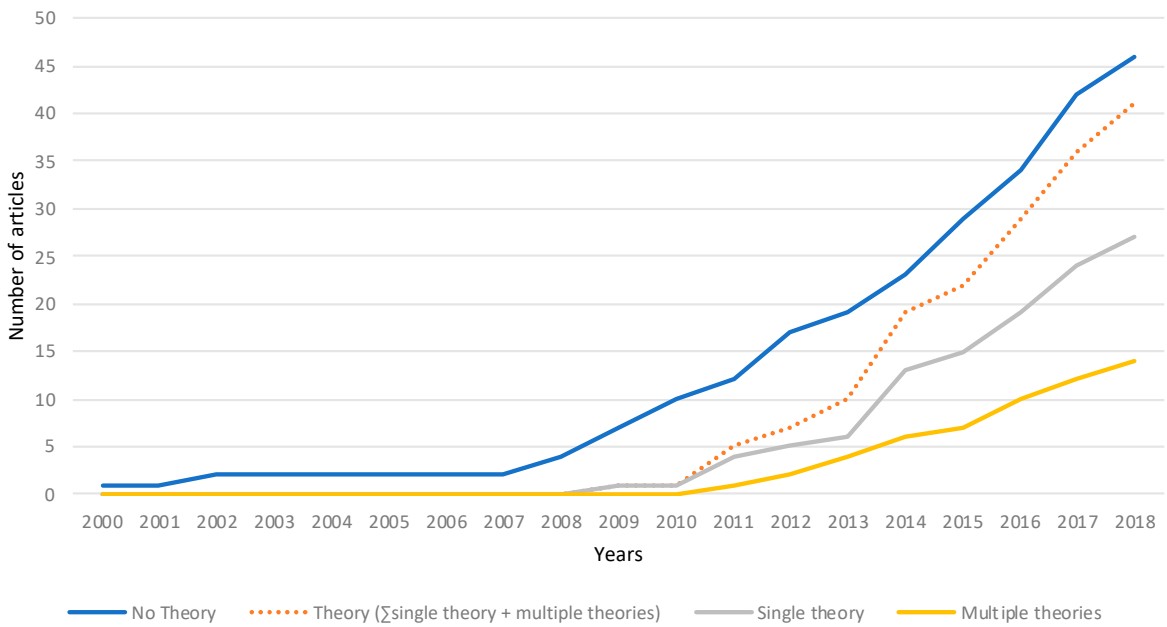

**Figure 9.** Development of "Theoretical lens" in articles over time.

The theoretical applications in the logistics and SCM journals is similar to that in the OM-journals [27], as the sampled articles are dominated by institutional theory [15,106,107], resource-based view including the natural resource-based view [63,74,76,95,121], transaction cost theory [9,12,56], stakeholder theory [73,75,104], and resource dependence theory [26,98]. Also, the findings of Touboulic and Walker [28] show similarities, which indicates that little change has taken place during the last five years (their sampling ended 2013), despite their call for new theoretical applications. The application of theoretical lenses indicates a perceived relevance of theories developed in other areas of research. One third of the theoretical articles draw upon multiple theoretical lenses [26,56,73,76,86,121]; illustrated by for example Hajmohammad and Vachon [86], taking a conceptual theory building approach, drawing upon resource dependence and agency theory to develop a framework of supplier sustainability risk management strategies. The resulting theoretical pluralism may bring increased benefits as the lens is not only a way of seeing, but also a way of "not seeing" [135].

The sample indicate that there is a link between the selection of theory and method applied. The favored theoretical lenses in the survey studies are transaction cost [9,56,101] and institutional theory [15,98,106], applied in isolation or together with other theoretical lenses. The studies where the lens of transaction cost or institutional theory are used display a focal firm (one actor) perspective on the IOSCI for sustainability. Relational theory [14,64] and coordination mechanisms [39,62] show multiple applications within the sampled case studies, applied in both one (focal dyad and focal supply chain) and two (dyad) actor studies. The conceptual articles favor the use of stakeholder theory [75,104,108], where most of the studies have the supply chain as the unit of analysis.

Only one [89] of the nine studies with network as UoA applies a theoretical lens (see Table 3). The use of theories is most common when supply chain is the UoA, and relatively speaking, dyadic studies show greater theory application than network studies. This may indicate that the applicability of the common inter-organizational theories is higher along supply chains (extending the UoA from dyad to triad) than in a network setting. A plausible explanation is the relatively higher inter-connectedness in relations between actors in a network—i.e., the relation a firm has with one customer is affected

by relationships formed with other customers. However, a customer relationship is naturally more decoupled from supplier relationships, which may complicate the application of theory in networks. In relation to empirical scope, there is no clear tendency related to theory application (see Table 2) as a large share of the studies with supply chain as the UoA are conceptual (see Table 3). As the research area is under development, future testing of the conceptual models could be expected, which may create a domination of studies with the supply chain as the empirical scope, i.e. a similar situation as for UoA.

**Table 3.** Theory use in relation to "Unit of analysis".

|  | Theoretical Lens | |
| --- | --- | --- |
| Unit of Analysis | No Theory | Theory |
| Dyad | 23 | 18 |
| Network | 8 | 1 |
| Supply chain | 15 | 22 |

### 3.7. Type of Sustainability Interaction

The importance of collaborative sustainability practices is a dominant theme in the literature [73,106,123,124]. However, by taking a broad point of departure with regard to different types of interaction and capturing relations spanning from arms-length [12,86,105] to long term close alliances or partnerships [56,86,89,92,95], it is shown that collaboration and control-based governance may take place at the same time in the same dyadic relationship [11,59,64,86,96,107]. This mixed approach can be exemplified through practices related to supplier monitoring, joint investment, and co-development [11,17,23,87], or by combining environmental demands and mutual adjustment of skills, knowledge, and norms [62]. A mixed approach is understood to suit relationships characterized by resource imbalance [64], and it has been shown that also control based interaction manages to deliver a positive outcome. Interaction for worker health and safety through codes of conduct is illustrated as a viable tool [12], especially when the most powerful actor enforces the code. Based on the sustainability dimensions, a wide variety of practices illustrating the IOSCI for sustainability are presented in Table 4.

**Table 4.** Examples of practices of IOSCI for sustainability.

| Environmental Practices | Social Practices |
| --- | --- |
| Sustainable sourcing [78,84] | |
| Supplier assessment [96,107] | |
| Green supplier selection [84,106,107] | Monitoring sustainability performance based on |
| Environmental requirements [62,68,70,82] | Codes of conduct [12,23,59,76,78] |
| Development of green programs [106] | Supplier self assessments [117] |
| Green knowledge exchange [80,94,98,109] | Suppliers report subcontracting arrangements [117] |
| Measurement tools for emissions [11,64] | Sustainable procurement [29,90] |
| Green technology investment [60,85] | Revenue sharing contracts [69] |
| Mutual adjustments and standardization of skills, | Coordinated pricing models [66] |
| knowledge and norms [62] | |
| Green packaging [29,39,110] | |

The type of sustainability interaction is largely understood to be dependent on the type of relations between supply chain actors. Several articles state that the nature of the relations sets the "scene" for what is a suitable type of IOSCI [63,68,73] where, e.g., lack of trust in the relationship is seen as a barrier to collaboration. Another point of view is that joint sustainability practice implementation can provide the potential to make the relation more strategic, i.e., the sustainability practice brings the possibility for more strategic and trust-based relationships [10,59,94,123]. Several articles show that certain types of relations are not necessarily prerequisites for supply chain sustainability practices, even though the type of relation may affect the outcome of such practices [12,30,63].

In line with the general literature on supply chain interaction, see, e.g., Ni and Sun [17] and Ralston et al., [33], trust and the power balance/power asymmetry [62,64,74,75] are two common aspects seen to influence the relationship and type of interaction formed. Power asymmetry often indicates the use of control mechanisms for transferring sustainability through the supply chain. This finding and that close alliances and partnerships build on more mutual responsibilities and joint sustainability practices, is a confirmation that IOSCI for sustainability is similar to other types of interaction. This implies that the general literature on coordination mechanisms is suited for research in this area. Joint dependency has been found to positively influence socially responsible supply chain management, while buyer dependence constrains it as the supplier can resist the implementation of sustainable practices [122].

## 4. Identified Knowledge Gaps

In order to identify research gaps this section draws upon the findings as presented in the previous section.

### 4.1. Sustainability Dimensions

The trend analysis identifies an increasing gap between the environmental and social dimensions, and it seems as if previous calls for research on the social dimension have not been adhered to. Environmental sustainability is however, proposed as the foundation for all sustainability, see e.g., Markman and Krause [137]. A larger focus on the environmental dimension might therefore be adequate and not at all a gap that must be closed. It is important that all three dimensions are addressed, and within the dimensions certain gaps are identified. Within the environmental dimension and despite areas other than $CO_2$ being addressed, other aspects such as resource consumption and the need for interaction for a circular economy could receive an increased research focus. Questions of social sustainability outside Asia and outside the product or materials suppliers is less addressed, why a research gap can be identified. Few articles present the complexity of managing the three sustainability dimensions simultaneously. The findings of Bals and Tate [138] support this finding. Very recently Xiao et al., [139] focused on the need for understanding the value of paradoxical sense-making in changes towards true sustainable SCM. There is room for more research applying a holistic view on sustainability as these dimensions are interlinked [127].

### 4.2. Unit of Analysis and Empirical Scope

Despite the importance of sustainability interaction beyond the dyad in order to truly gain sustainability performance advantages [29,30,32] and calls for SSCM research with supply chains as the UoA and data collection at multiple actors [31,32], a (buyer) focal firm emphasis remains. The bulk of studies were published in 2013 and later, yet few display collection of data from three or more interlinked actors, i.e., few studies reflect a true supply chain, or network. Investigating IOSCI for sustainability from the perspectives of all actors involved could provide new or deeper insights into the intricacies of such an interaction. This review, however, shows the opposite trend: the gap between studies with a one-actor perspective and those with multiple-actor perspectives is increasing.

Unlike previous SLRs within the same and adjacent areas, a relatively high share of UoA beyond the dyad is displayed. This is promising as in order to understand the level of sustainability of the supply it is important to look beyond one supply chain interface or one relation. A clear gap in relation to UoA has not been perceived. However, a research gap and a continued call for research with broad empirical scopes is issued based on the findings of this review.

### 4.3. Methods Applied

The logistics and SCM journals do not display the same research gaps as the OM journals [27]. Instead, a large share of the articles displays qualitative (multiple) case studies, relevant when studying complex phenomena [129]. However, action research—a participatory research process between

researchers and practitioners with a goal to improve a situation and hence offering clear managerial relevance to SCM research [140,141]—is perhaps an even better suited method to understand and improve messy and complex phenomena [142]. This review displays few such studies, thus granting a relevant research gap. This review also identifies a clear research gap in terms of few longitudinal studies. Longitudinal research can bring new insights into how IOSCI for sustainability evolves and unfolds over time [143,144]. The current research gives "what is" snapshots; however, it is not easy to assess when and in what ways, e.g., close relationships, is a precondition for the sustainability practices implemented, or if the sustainability practices are the foundation for close relationships. A deeper understanding of those relationships could be provided by longitudinal studies.

### 4.4. Industry and Geographical Area

Few studies address service industries such as healthcare, transportation and warehousing. As logistics and supply chain management research is targeted, and the transportation industry impacts environmental sustainability the most [145], it is surprising that only four of the sampled articles provide empirical insights into IOSCI involving LSPs. This finding points to a research gap regarding transportation. Also, research involving wholesalers is scarce and is a research gap to fill, though their impact on sustainability may be lower.

Studies of IOSCI for sustainability spread over the globe, however with a clear European and Asian dominance. An almost exponential increase in Asian studies—well above that of, e.g., European and North American—is displayed. If this trend continues, a bias towards Asian research is inevitable, which is why a plausible future research gap of "non-Asian studies" can be identified. As regions differ in terms of external drivers such as social expectations, regulatory requirements, industry norms [15,106,107], industry maturity, and culture [16], a research bias toward a certain region may skew the picture of IOSCI for sustainability. Thus, studies from less represented regions such as South America, Africa, or even Oceania may give interesting and novel insights, a complementary perspective and an understanding of external and internal drivers and barriers in that context. Therefore context-spanning research may render a richer, more nuanced understanding of interaction for sustainability.

Despite many supply chains being international, the majority of the studies address IOSCI within one country (or region), this opens for more studies crossing country and/or regional boundaries; a research gap which can be addressed in future research. Those studies could offer insights into the sustainability practices and interactions focused on, the complexities involved when western buyer firms pursue sustainability in emerging markets, as sustainability and other business aims may be in conflict (see e.g., Xiao et al., [139] for an illustration), as well as drivers of IOSCI appearing in supply chains that stretch over cultural and geographical distances. Finally, in relation to empirical scope, the studies of interlinked supply chains are based on case study methodology and are primarily from Asia and Europe; North America—likely as a reflection of the preferred research paradigm [142]—displays only one of these. Research on IOSCI for sustainability in interlinked supply chains, in a North American context specifically, can be viewed as a research gap to fill.

### 4.5. Theoretical Lens

As the gap between non-theoretical and theoretical articles is narrowing, the use of theories as such does not indicate a clear research gap. Based on the findings of Touboulic and Walker [28], a wider application of, e.g., network theory and agency theory, could have been expected. The use of those lenses could provide deeper insights into, e.g., incentive mechanisms (agency) for IOSCI for sustainability or the diffusion of sustainability practices deeper into the supplier network. The relative lack of certain theoretical lenses can be perceived as a research gap. In terms of theory building, a number of articles display such an aim [10,39,59,67,68,73,74,76] and no clear research gap can be identified. Regardless, this is a development to be further encouraged. In studies with the network UoA,

the application of theory is lacking, i.e., a research gap can be identified, and testing the applicability of different theoretical lenses in this setting may therefore be a suitable future endeavor.

### 4.6. Types of Interaction

The need for deepened understanding of how e.g., power asymmetries and trust influence the interaction for sustainability, and what sustainability initiatives are suitable for implementation emerges as a research gap. Identifying barriers and enablers, and understanding the "direction of the relation" between IOSCI for sustainability and the relationships formed, can be fruitful in order to give clearer indications of how relationship types affect IOSCI, or vice versa.

Mostly positive, yet also negative associations between IOSCI for sustainability and (financial) outcomes are identified; accentuating importance of the sustainability 'business case' and the treatment of investments in sustainability as any other strategic investment. Focusing on the sustainability business case and profitable ways of securing sustainability on firm, dyad, network, or supply chain level are relevant areas for further research.

## 5. Conclusions and Future Research Agenda

This review was undertaken by investigating and describing the articles in view of, e.g., sustainability dimensions, applied research methods, types of inter-organizational supply chain interaction, and their development over time. Literature reviews in adjacent areas display similarities regarding, e.g., theory applications [27,28]. The study also addresses the white space regarding SLRs in SSCM put forward by Carter and Washispack [125], by investigating relationships among e.g., empirical scope and theoretical lens and methods applied. Furthermore, by doing so identifying the need for further theory development and use regarding e.g., network studies where theory is rarely applied.

Through its focus on development over time, this review contributes to knowledge by illuminating that the gap between the social and environmental dimensions is growing rather than closing. It also identifies a nearly exponential increase in studies of IOSCI for sustainability in Asia which, if it continues, will have implications for how IOSCI is understood. The trend analysis further identifies a relative increase in studies with a one-actor or focal firm perspective. Through the specific focus on unit of analysis (UoA) and empirical scope, one contribution to literature lies in the identification of eight empirical scopes—"true" or "focal firm" dyad, supply chain and network, as well as dyads and supply chains "not interlinked." This study draws attention to potential links between theory application and UoA, where the network studies lack the application of theoretical lenses, and the choice of method and empirical scope where the case study methodology seems to favor a broader empirical scope. Limited research where the IOSCI stretches beyond the dyad (UoA) with data collection from all involved actors (empirical scope) has been published. The majority of studies have a focal firm perspective, showing only one perspective on the IOSCI for sustainability. One conclusion is therefore that the knowledge about how IOSCI is experienced by multiple actors in interlinked (or true) supply chains is scarce, similar to, e.g., the findings proposed by Carter and Easton [31]. In relation to OM journals, the logistics and SCM journals display more case studies and qualitative methods used. Besides a contribution to research, this study has managerial implications. By taking a broader perspective on different types of interaction, it becomes clear that interaction for sustainability is not, and should not, be limited to strategic relationships. From a managerial point of view, this gives support for investments in also more operational forms of interaction (cooperation). Practitioners can also benefit from the proposed research agenda, where a focus on more participatory research studies with immediate dissemination of research results to industry can take place. One of the main contributions to literature is the proposed research agenda based on gaps or less covered aspects (see Table 5), laying the foundation for relevant future studies. On a general note certain geographical areas are clearly under-represented (South America and Africa, though possibly included in studies of 'suppliers in developing countries' as well as Australia/Oceania); certain other regions are under-represented related to the application of specific research methods and study objects (e.g., North America related

to qualitative case studies and studies of true supply chains or networks). On a more specific note, the following aspects are proposed to be included in a future research agenda:

**Table 5.** A proposed future research agenda.

| Aspect | Description | Example Research Topics | Suggested Methodology |
|---|---|---|---|
| *Research addressing the business case of sustainability.* | The identified recent trend combining the environmental and economic dimensions indicates that sustainability may be a business opportunity and very well pay off. However, sustainability initiatives may need to be evaluated and assessed in relation to other strategic or tactical initiatives. | For the full integration of sustainability into SCM and for the development of practice, investigations of how economic, social and environmental sustainability performance is quantified and traded off in firm internal, as well as inter-organizational business cases is an important avenue for further research. | Application of a case study methodology in true dyads, supply chains or networks to provide deepened understanding of e.g., how firms estimate and build business cases around inter-organizational sustainability initiatives. Participatory research could for example be applied to refine the business case in action, and to go from "what is said" to "what is actually done". |
| *Research addressing the effects of supply chain position on IOSCI for sustainability.* | Most research target materials- and product suppliers. However, sustainability work and related performance outcomes differ at different positions in the supply chain [146], likewise does the impact of supply chain integrity on the SSCM of firms [147]. Relatedly, IOSCI and relationships may differ in upstream and downstream relationships. Larger focus on e.g., retailers in future research is needed. | Further investigations are proposed to understand how relationships and interaction for sustainability differ along supply chains, and what sustainability dimensions and practices are adopted in upstream vs downstream relationships; including the reasons for the potentially different approaches. | Survey studies of true, or not interlinked dyads or supply chains, analyzing the focused sustainability dimension and inter-organizational sustainability practices based on supply chain positions, can provide further understanding of this aspect. |
| *Research addressing the involved actors' experiences and perceptions of IOSCI.* | Involved actors may have different perceptions and experiences of IOSCI; not clearly captured in extant research. Research to capture the differing views of diverse actors (e.g., top management team-managers-first line workers; buyer/supplier representatives) in real supply chains or networks would be valuable from a managerial perspective. Especially as this is the real setting for IOSCI in the business world. | To broaden the understanding, the different perspectives of actors need to be investigated. Actors, based on e.g., position, age or gender may perceive the interaction differently. See e.g., Kumar and Paraskevas [148] noting increasingly proactive environmental strategies for firms with female representation in top management teams. In order to make research increasingly relevant for practitioners, also this kind of nuanced understanding is important. | In-depth interview studies and participant observations are recommended as these can provide a deeper understanding of actors' perceptions as well as how the actual IOSCI plays out. Participatory research increases the possibility to speed up the dissemination process to the industry. Studies based on true dyads, supply chains and/or networks are recommended. |
| *Social and environmental sustainability research addressing also service suppliers.* | Issues related to working conditions, safety, and wages are not isolated to product and materials suppliers, neither is environmental footprint. In order to create a truly socially and environmentally sustainable product offering also the, often invisible, service suppliers must be sustainable. A focus on LSPs and other actors in the transport industry provides the opportunity for research covering a larger share of the supply chain's environmental and social footprint. | To reach a more holistic understanding also interactions with service suppliers needs to be investigated. Research is proposed to involve e.g., applied social sustainability performance indicators in shipper-transport-provider relationships; how transport suppliers interact with shippers using social sustainability performance indicators as sales arguments; how transport service providers interact with shippers for reduced environmental supply chain footprint. How the interaction is played out in tiered supply chains is another important aspect to investigate. | Studies could be carried out with different methodologies, such as sustainability report scans, case studies and surveys. |
| *Research designed in order to understand how IOSCI and relations evolve over time.* | In order to understand the prerequisites for IOSCI and how it evolves or unfolds over time, an understanding of how relations affect and depend on IOSCI is needed. For example, cause-and-effect relationships could be investigated. | Previous research has to some extent investigated the relationships between relation and IOSCI; future research could firm those findings by investigating the mechanisms affecting them. | Longitudinal studies of evolving IOSCI and relations in true dyads are proposed. |

**Author Contributions:** Conceptualization, V.S.Ü., M.B. and H.F; methodology, M.B, V.S.Ü. and H.F; software, V.S.Ü.; validation, H.F, N.S., V.S.Ü. and M.B.; formal analysis, V.S.Ü., M.B. and H.F.; investigation, V.S.Ü., M.B. and H.F.; writing—original draft preparation, V.S.Ü., M.B. and H.F.; writing—review and editing, M.B.; V.S.Ü.; N.S. and H.F.; visualization, N.S. and V.S.Ü.; supervision, M.B. and H.F.; project administration, M.B.; funding acquisition, M.B. and H.F.

**Funding:** This research was funded by The Kamprad Family Foundation for Entrepreneurship; Research and Charity, grant number 20160039.

**Acknowledgments:** The authors acknowledge the support from the librarian Per Eriksson at Linköping University in conducting the literature searches.

**Conflicts of Interest:** The authors declare no conflict of interest. The funders had no role in the design of the study; in the collection, analyses, or interpretation of data; in the writing of the manuscript, or in the decision to publish the results.

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
