# Peer review of "Inter-Organizational Supply Chain Interaction for Sustainability: A Systematic Literature Review"

_sustainability, doi:10.3390/su11195488_

Round 1
Reviewer 1 Report
The purpose of this paper is to advance the understanding of the logistics and supply chain management literature on IOSCI for sustainability by carrying out a systematic literature review (SLR) The paper attempt to synthesize the existing knowledge and ongoing trends, and to propose a research agenda. I thank the authors for giving me the opportunity to review their manuscript. The comments are very critical, but please accept that my comments are intended to help and improve the quality of your research. While the manuscript potentially contain some new insights to the field of logistics and SCM, there are way too many issues for it to be publishable in its current form. Some key limitations/issues that need to be addressed are described below. I sincerely hope you find the comments as constructive.
The core issue of the paper is that the SLR is “limited to journals with a clear and narrow focus on logistics and supply chain management”. According to the authors the reasoning for this limitation is “It is relevant to target not only sustainability journals; rather to bring in knowledge to sustainability journals from other areas.” I argue that this is a fatal mistake and that valuable papers from the sustainability field is omitted that would significantly impact the findings of the SLR. As a scholar I am surprised that the paper is submitted to a journal in the field of Sustainability which contributions seems to be excluded.
Another methodological issue of the paper is the inclusion and exclusion criteria. Stated by the authors, SLR allows the maximization of replicability. However, the inclusion and exclusion criteria is not explicitly stated. Without these it is impossible for other scholars to replicate the study. The inclusion and exclusion criteria need to be formulated in precise manner. It is also strongly suggested to further develop Figure 1 to illustrate how many paper was excluded based on each criterion. An excellent element in the SLR is its reliability - achieved by the author’s joint rigorous process – well done concerning this quality aspect.
In the introduction section the authors relate to previous published SLRs on sustainable supply chain interactions, which point out that there is a need for more logistics and SCM research targeting social sustainability as the environmental dimension is overrepresented. Based on this, the authors argue that it is unclear to what extent the research has been able to narrow this gap or if the gap is expanded, thus arguing the need for another SLR on the topic. This reasoning is unclear – what research or scholar say that there should not be a gap, or that research should narrow this gap – maybe there should be a gap? Does it have to be “equal” amount of papers between these two dimensions? If so, why? In overall, the motivation “WHY” another SLR is needed is weak and need to be strengthened. What original/novel contribution does this SLR paper provide compared to the existing ones? This is especially important in this paper as there are a handful of SLR on the topic published relatively recent – with similar findings.
I argue that the analysis/results/synthesizes of the SLR have not reached its full potential and lack an impactful story. In section three “Results” the paper need to move beyond simple descriptions and pinpoint core insights and results/findings from the SLR. For example in section “3.1 Source and date of publication” contain just description and no interpretation or explanation why this is of interest. Another example is section “3.2 UoA and empirical scope” where the three groups of UoA is described in terms like “the largest and fastest growing”, “the runner-up”, and “at much slower growth”. Looking at the graph in figure 3, it is not about being the first, second or last. I would argue that the interesting thing is that dyad and supply chain as UoA are similar in growth while network as UoA is lagging behind. If the explanation of this “lagging behind” is related to survey/methodology that data needs to be shown (e.g. in a table) in order to adequately tie together the findings/elements of the paper. To do this, it would be reasonable to present “methods applied” after UoA, and not “empirical scope”. In the same manner it seems that “UoA” is strongly associated with “Theoretical lens”, while “empirical scope” is associated with “industry”. One suggestion would be to integrate section 3 and section 4 in order to present an impactful story. Having these sections separate make it very difficult for the reader to follow the arguments and reasoning. To sum up, each sub-section in the results section need to be carefully looked at and reflected upon to make sure that each sub-section and paragraph is presented clearly and analyzed appropriately.
This leads to another important issue; Sustainability need to be woven into the text throughout section 3. The first six pages of section three (i.e. Result) primarily focus on IOSCI and only occasionally relates to sustainability. It is in sub-section 3.6 Sustainability dimensions, it is finally discussed. However, the discussion revolves around “trends” as number of papers and not the “trends” in literature content. Only examples of content is described, while a thorough synthesis of all papers content is expected to be presented. Similar criticisms can be applied to sub-section 3.7 Type of interaction. The table 4 of practices is interesting, but not only examples – a synthesis of all practices found and references to that literature.
Finally, a weakness of the paper is the quality of communication. The paper is not very well written and the quality of communication is below acceptable standard. The quality of English used and sentence structure is good though. However, large parts of the text is very wordy (resulting in a lengthy paper) and the scientific writing (e.g. overall paragraph structure with topic sentences and clear reasoning, descriptions and explanations) are not satisfactory, resulting in a lack of readability. Moreover, there are also repetition in the paper indicating that the authors have not carefully addressed the writing of the paper as a whole. This results in a weakness in communicating an impactful story that makes the study and its contribution clear and trustworthy. The authors are suggested to carefully reassess the structure and writing of each section and paragraph of the manuscript. Some additional suggestions:
Use tables and figure appropriately in order to not cause confusion for the reader. For example; Why have ID for the included articles in table 1? What value do Figure 5? Where to have table 2, which includes new data, as well as data not used/discusses, and later introducing table 3 with theoretical lens in other dimensions? Add more complete and comprehensive tables with data! In current version of the manuscript many parts of the SLR data are described in the text, instead of displayed in a table. The title of the paper “IOSCI for sustainability” implies something different than “IOSCI and sustainability”, but seems not to be found when reading the paper methodology or its results.
Author Response
Thank you for your constructive comments, it has helped us to develop the paper further strengthening e.g. argumentation and providing clearer logic and connection to sustainability. You find our comments in the attached file.
Again thanks, we see that you put in a lot of effort in this review!
Kind regards,
The authors

Reviewer 2 Report
Summary of the review:
The article is well-structured. The English language is proper, maybe some details can be improved, but it is clear and easy to understand for the reader.
The whole method, as well as the research procedure, process and results are described well. Also, the value of the research results for the studied field is high and can be the basis for further, more empirical studies.
Abstract:
Line 9-10
One sentence should be added, why this research is so important. Advancing knowledge is not so convincing. Some gaps, e.g. knowledge gap, should be mentioned here, shortly. This will attract the paper for the potential audience.
Line 14
"the minimization of researcher bias" is not clear.
Introduction:
Line 35-36
"see e.g. [9,10], to more operational, co36
operational forms see e.g. [11,12]." - this should be revised. The same procedure for the whole document because similar mistakes are made in the next paragraphs, e.g. Line 38. Please revise it in the whole text.
Method:
Line 101
I would prefer to see the figure with all the steps of the research. Figure 1 is not complete to view the whole process. Please remove Figure 1 and replace it with the new figure about the whole process, starting from keywords identification/establishing.
Line 130-131
"Economic sustainability was not an inclusion criterion on its own." It should be explained/justified, why this third dimension of sustainability was not included in the research
Results:
I have no comments on this section. This section was made properly and doesn't need any changes.
Identified knowledge gaps:
I have no comments on this section. This section was made properly and doesn't need any changes.
Conclusions:
Only one technical comment - check if all commas are included in the text. I found some few places where they should be placed. The text is correct, clear, well-structured. Table 5 is a good summary of this section.
General technical comment:
Please remove unnecessary commas like "e.g.," in the abstract and in the whole text.
Author Response
Thank you fore your positive feedback on our paper. We have strived to improve the paper even further by addressing the remaining of your concerns. See the comments in the attached file.
Kind regards,
The authors

Round 2
Reviewer 1 Report
Very well done. The authors have addressed all main issues and significantly improved the paper. I only have one minor issue. That is, the paper is still “wordy”. Just go through the text (sentences and paragraphs) in order to modify, condense, and delete text that will shorten the paper without changing the content. Addressing this will make the paper communication quality even better. Once again, well done!
Author Response
Dear editors,
We have addressed the issue regarding "wordiness", rewritten wordy sentences without changing the content. Hope you approve of our changes made.
Best regards,
Maria Björklund & Coauthors